# Deficiency in PHD2-mediated hydroxylation of HIF2α underlies Pacak-Zhuang syndrome
Fraser G. Ferens [1,2], Cassandra C. Taber[1], Sarah Stuart[1,2], Mia Hubert[1], Daniel Tarade[1], Jeffrey E. Lee [1] & Michael Ohh [1,2] ✉

Pacak-Zhuang syndrome is caused by mutations in the *EPAS1* gene, which encodes for one of the three hypoxia-inducible factor alpha (HIFα) paralogs HIF2α and is associated with defined but varied phenotypic presentations including neuroendocrine tumors and polycythemia. However, the mechanisms underlying the complex genotype-phenotype correlations remain incompletely understood. Here, we devised a quantitative method for determining the dissociation constant ($K_d$) of the HIF2α peptides containing disease-associated mutations and the catalytic domain of prolyl-hydroxylase (PHD2) using microscale thermophoresis (MST) and showed that neuroendocrine-associated Class 1 HIF2α mutants have distinctly higher $K_d$ than the exclusively polycythemia-associated Class 2 HIF2α mutants. Based on the co-crystal structure of PHD2/HIF2α peptide complex at 1.8 Å resolution, we showed that the Class 1 mutated residues are localized to the critical interface between HIF2α and PHD2, adjacent to the PHD2 active catalytic site, while Class 2 mutated residues are localized to the more flexible region of HIF2α that makes less contact with PHD2. Concordantly, Class 1 mutations were found to significantly increase HIF2α-mediated transcriptional activation in cellulo compared to Class 2 counterparts. These results reveal a structural mechanism in which the strength of the interaction between HIF2α and PHD2 is at the root of the general genotype-phenotype correlations observed in Pacak-Zhuang syndrome.

Missense mutations in the *EPAS1* gene have been shown to cause neuroendocrine tumors with or without polycythemia or polycythemia in isolation[1–4]. The *EPAS1* gene encodes for the hypoxia-inducible factor 2 alpha (HIF2α), an essential component of the cellular oxygen-sensing pathway in humans. The disease manifestations caused by mutations in *EPAS1* have been referred to as Pacak–Zhuang syndrome or more recently as collective HIF2-driven disease, which can be segregated into two distinct disease classes caused by unique subsets of mutations[4,5]. Class 1 disease is associated with various combinations of pheochromocytoma, paraganglioma, and somatostatinoma with or without polycythemia while Class 2 disease is associated with the exclusive development of polycythemia and is also known as Familial Erythrocytosis Type 4[4]. Class 1 disease can be further subcategorized into Class 1a, characterized by the presence of pheochromocytoma and paraganglioma (PPGL), somatostatinoma, and polycythemia; Class 1b, characterized by the presence of PPGL and polycythemia; and Class 1c, characterized by the presence of only PPGL (Table 1)[4]. Notably, disease-causing *EPAS1* mutations regardless of Class have been shown to upregulate HIF2α expression level[2,6].

In vertebrates, HIF2α is one of three oxygen-labile HIFα paralogs that act as the master regulators of the hypoxia transcriptional response. Under oxygenated conditions, HIFα subunits are hydroxylated by prolyl-hydroxylase enzymes (PHDs) at two conserved proline residues within the oxygen-dependent degradation domain (ODD). This hydroxylation reaction uses molecular dioxygen as a substrate, defining PHDs as oxygen sensors[7]. Hydroxylated HIFα is recognized by the von Hippel-Lindau tumor suppressor protein (pVHL), the substrate receptor of an E3 ubiquitin ligase comprised of Elongin B, Elongin C, Cullin-2, and Rbx1 that poly-ubiquitylates hydroxylated HIFα subunits targeting them for destruction by the 26S proteasome[7]. Under low intracellular concentrations of molecular dioxygen, HIFα remains unhydroxylated and escapes degradation initiated

[1]Department of Laboratory Medicine & Pathobiology, Faculty of Medicine, University of Toronto, 1 King's College Circle, Toronto, ON M5S 1A8, Canada. [2]Department of Biochemistry, Faculty of Medicine, University of Toronto, 661 University Avenue, Toronto, ON M5G 1M1, Canada. ✉e-mail: michael.ohh@utoronto.ca

**Table 1 | Pacak–Zhuang syndrome classes, manifestation, and mutation positions**

| Pacak–Zhuang syndrome class (subclass) | Disease manifestations |
|---|---|
| 1(a) | PPGL, Somatostatinoma and Polycythemia |
| 1(b) | PPGL[1] and Polycythemia |
| 1(c) | PPGL |
| 2 | Polycythemia |
| **Amino Acids positions of Pacak–Zhuang syndrome missense mutations**[a] | |
| ...511                                                    550...<br>AKDQCSTQTDFNELDLETLAPYIPMDGEDFQLSPICPEER<br>                                       * | |

*PPGL* pheochromocytoma and paraganglioma.
[a]Underlined residues comprise the most frequently mutated region[4,45], positions mutated in Class 1 disease are colored red, positions mutated in Class 2 disease are colored blue and purple positions denote Class 1 or Class 2 disease depending on the mutation. Hydroxylated proline (P531) is indicated by an asterisk underneath the residue.

from pVHL recognition. The stabilized HIFα forms a heterodimeric transcription factor with the constitutively expressed HIFβ subunit and translocates to the nucleus where the heterodimeric HIF transactivates numerous hypoxia-responsive genes to trigger cellular adaptation to compromised oxygen availability[8–10].

The two conserved prolines of HIF2α targeted for hydroxylation by PHDs and recognized by pVHL are P405 and P531[8]. Pacak–Zhuang syndrome mutations are concentrated in the HIF2α sequence to the residues in proximity to P531 within the C-terminal ODD (CODD) region (Table 1)[4]. Mutations in the HIF2α-CODD region likely disrupt the interaction with either PHDs or pVHL resulting in the stabilization of HIF2α above wild-type (WT) levels under normoxic conditions[2,4,6]. This increase in HIF2α level consequently leads to an inappropriate hypoxic or pseudo-hypoxic transcriptional response, which ultimately leads to the observed disease phenotypes through changes in the expression of downstream HIF-responsive genes such as erythropoietin (*EPO*), vascular endothelial growth factor (*VEGF*) and stem cell pluripotency genes[9]. This potential disease mechanism is reminiscent of VHL disease, which disrupts the negative regulation of HIFα paralogs through mutations in the *VHL* gene and is associated with the development of PPGL and polycythemia among other cancers such as central nervous system haemangioblastoma and clear-cell renal cell carcinoma[5,11,12].

In an effort to elucidate the mechanism(s) underlying the emerging genotype-phenotype correlations in Pacak–Zhuang syndrome, we previously aggregated 66 reported cases of diseases caused by *EPAS1* mutations and showed that the Class 1 and Class 2 diseases were associated with distinct sets of mutations and that most Class 1 mutations markedly reduced the binding affinity of prolyl-hydroxylated HIF2α peptides to pVHL in comparison to Class 2 mutations[4]. These findings supported the notion that the extent of HIF2α stabilization can be predictive of disease class outcome. However, certain disease-causing mutations did not adhere to this general model as some mutations did not diminish pVHL affinity for HIF2α[4]. Here, we explored the impact of class-defining mutations on the upstream HIF2α–PHD2 interaction with the supposition that such insight might provide a more comprehensive model of pathogenesis underlying Pacak–Zhuang syndrome.

## Results

### Indirect hydroxylation assay does not sufficiently distinguish between Class 1 and Class 2 mutants

We previously showed that synthetic HIF2α peptides that have been in vitro hydroxylated via purified recombinant PHD2 bind pVHL and that Class 2 mutants were more readily hydroxylated than Class 1 mutants as indicated by stronger binding to pVHL[4]. Here, we examined an expanded set of clinically relevant HIF2α-CODD peptides (Supplementary Table 1) to establish a comprehensive data set. We showed that Class 1 mutant peptides co-precipitated pVHL modestly or negligibly following in vitro PHD2-mediated hydroxylation, which infers weak levels of hydroxylation (Fig. 1a and Supplementary Fig. 1). In contrast, Class 2 mutants displayed variation with some mutants producing modest or negligible signals while others showing near-WT signals in comparison to the WT HIF2α-CODD peptide (Fig. 1a and Supplementary Fig. 1). These results demonstrate that the indirect hydroxylation assay does not capture a reliable distinction between disease classes as several Class 1 and Class 2 mutants were indistinguishable. This likely arises from a number of confounding variables in these types of experiments such as batch variability in in vitro transcribed HA-pVHL and the qualitative nature of Western blots.

### PHD2 has lower affinity towards Class 1 mutants than Class 2 mutants

The indirect or proxy hydroxylation assay is a qualitative approach that does not provide an accurate quantitative comparison of the effect each disease-associated mutation has on hydroxylation or the interaction with PHD2 or pVHL. To determine if there is a direct effect of HIF2α mutations on PHD2 binding, we devised a quantitative method for determining the dissociation constant ($K_d$) of the HIF2α-CODD peptides and the catalytic domain of PHD2 using microscale thermophoresis (MST)[13,14] in the presence of the PHD inhibitor n-oxalylglycine (NOG) to inhibit reaction turnover. First, we assessed the potential of the method by examining the interaction between PHD2 and HIF2α-CODD WT peptide or the synthetically hydroxylated WT HIF2α-CODD-OH peptide as a negative control. The $K_d$ of HIF2α-CODD WT peptide and PHD2 was 34 μM while HIF2α-CODD-OH peptide did not bind to PHD2 as expected (Fig. 1b and Table 2)[15]. After confirming the method was suitable for accurately determining the $K_d$ of the HIF2α-CODD/PHD2 interaction and that the method could distinguish between bound and unbound peptides, we examined the effect of the HIF2α mutations on the $K_d$ of the interaction. All Class 1 and Class 2 mutant peptides were observed to have $K_d$ values higher (i.e., weaker interaction) than the WT peptide (Fig. 1c and Table 2). Notably, the $K_d$ values were stratified by Pacak-Zhuang syndrome class with all tested Class 1 mutations either having higher $K_d$ than any Class 2 mutation (≥345 μM, ≥10× WT) or not binding to PHD2 (Fig. 1c and Table 2). The majority of Class 2 mutations had $K_d$ values within a narrow range (150–230 μM; 4.4–6.5× WT) with Y532H and G537W having $K_d$ values of 41 μM (1.2× WT) and 83 μM (2.4× WT), respectively. Interestingly, the Y532C peptide, a Class 1 mutation at the same position as the Class 2 Y532H, had a $K_d$ value above all Class 2 mutations (345 μM) indicating that the property of the substituted amino acid is important at this position for binding PHD2. Similarly, the G537R Class 2 mutation, which falls in the same position as the Class 2 G537W mutation, had a $K_d$ value (175 μM) within the range of the majority

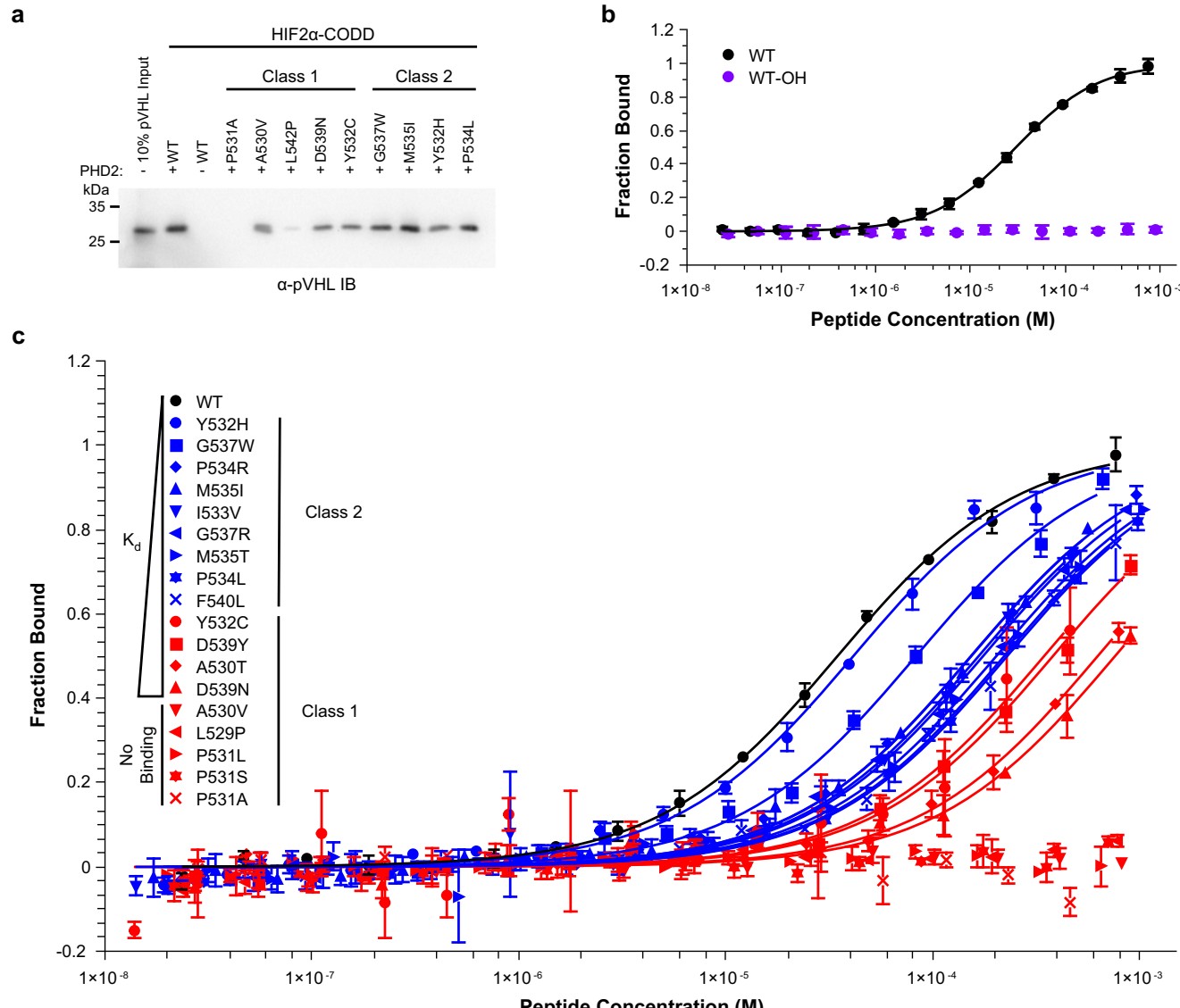

**Fig. 1 | Mutant HIF2α-CODD peptide affinity to PHD2 is correlated with Pacak–Zhuang syndrome class. a** Indirect hydroxylation assays of HIF2α-CODD peptides representing Class 1 and Class 2 Pacak-Zhuang syndrome-associated mutations. Immobilized HIF2α-CODD peptides were hydroxylated by PHD2. HA-pVHL was pulled down by hydroxylated peptides and an α-HA-pVHL Western blot was used to assess the extent of hydroxylation. **b** MST-binding curves of WT HIF2α-CODD peptide (WT) and synthetically hydroxylated HIF2α-CODD peptide (WT-OH) to PHD2. Solid lines represent the $K_d$ function fit to the experimental data (solid circles). **c** MST binding curves of WT, Class 1 and Class 2 mutant HIF2α-CODD peptides to PHD2. Solid lines represent individual $K_d$ function fit to the measured data points (solid shapes). Data and function fit are colored by Pacak–Zhuang syndrome Class; Class 1 is red, Class 2 is blue and WT is black. Individual non-normalized PHD2 MST binding curves for each peptide can be found in Supplementary Fig. 2. In panels **b** and **c**, error bars represent the standard deviation of triplicate measurements.

of Class 2 mutants analyzed. Two mutant peptides, L542P (Class 1) and D539E (Class 2), were observed to cause aggregation of PHD2 in MST experiments and hence binding assessment or dissociation constants could not be determined (Table 2 and Supplementary Fig. 3). These results suggest that PHD2 affinity for mutant HIF2α-CODD peptides is correlated to disease class and may serve as the basis for predicting the broad disease phenotype of novel mutations within the HIF2α-CODD region.

## PHD2 has lower affinity for HIF2α-CODD than HIF1α-CODD
We aimed to use the structural insight of the PHD2/HIF2α-CODD interface to better understand the molecular mechanisms for the reduced affinity of PHD2 for each of the disease-associated HIF2α-CODD mutants relative to the WT peptide. However, there was no available structure of the PHD2/HIF2a-CODD complex at the time of our investigation. We initially thought to use the previously solved PHD2/HIF1α-CODD complex structures[15–17] as

the sequences of HIF1α-CODD and HIF2α-CODD are very similar; there are in fact only 4 amino acid differences between the human HIF1α-CODD and HIF2α-CODD regions (Fig. 2a). Of these 4 amino acids, we predicted that HIF2α T528 (corresponding to HIF1α M561) and HIF2α G537 (which is an insertion relative to HIF1α) to have the largest impact on the binding interface due to the nature of these differences (Fig. 2a). In order to assess the usefulness of a PHD2/HIF1α-CODD structure for the interpretation of our binding data, we sought first to determine the influence of each of these two differences between the two HIFα-CODD sites on the interaction with PHD2. We examined HIF2α-CODD peptides altered at each position independently (T528M and ΔG537) by MST. We observed that PHD2 had approximately 3x higher affinity for HIF1α-CODD than HIF2α-CODD (Fig. 2b and Table 2), which was consistent with a previous examination of the $K_m$ values for the hydroxylation reactions of each peptide by PHD2[18]. We next determined that PHD2 has an intermediate affinity for HIF2α-

**Table 2 | MST-determined PHD2/HIFα-CODD complex dissociation constants**

| Non-disease associated peptides | $K_d$ (μM) | | |
|---|---|---|---|
| HIF1α WT | 13 ± 1 | | |
| HIF2α WT | 34 ± 2 | | |
| HIF2α WT-OH | No Binding | | |
| HIF2α ΔG537 | 13 ± 2 | | |
| HIF2α T528M | 23 ± 2 | | |
| **Pacak–Zhuang syndrome mutant peptides** | **$K_d$ (μM)** | **Pacak–Zhuang syndrome class(-sub-class)** | **Number of reported cases** |
| HIF2α Y532H | 41 ± 5[a] | 2 | 2[46] |
| HIF2α G537W | 83 ± 9 | 2 | 3[1,47] |
| HIF2α P534R | 153 ± 10 | 2 | 1[48] |
| HIF2α M535I | 154 ± 17 | 2 | 2[49] |
| HIF2α I533V | 167 ± 29 | 2 | 3[45,50] |
| HIF2α G537R | 175 ± 20 | 2 | 24[45–48,50–52] |
| HIF2α M535T | 203 ± 25 | 2 | 7[45,48,53,54] |
| HIF2α P534L | 221 ± 19 | 2 | 1[6] |
| HIF2α F540L | 229 ± 25 | 2 | 4[45,48] |
| HIF2α Y532C | 345 ± 247 | 2 | 1[45] |
| | | 1(a) | 1[55] |
| | | 1(c) | 2[56,57] |
| HIF2α D539Y | 381 ± 50 | 1(c) | 1[58] |
| HIF2α A530T | 627 ± 126 | 1(a) | 1[55] |
| | | 1(b) | 1[58] |
| HIF2α D539N | 760 ± 274 | 1(a) | 1[59] |
| | | 1(b) | 2[59,60] |
| HIF2α A530V | No binding | 1(a) | 1[55] |
| | | 1(b) | 4[57,60,61] |
| | | 1(c) | 2[58,62] |
| HIF2α L529P | No binding | 1(a) | 2[55,63] |
| HIF2α P531L | No binding | 1(b) | 1[58] |
| | | 1(c) | 2[56,64] |
| HIF2α P531S | No binding | 1(b) | 6[57–60,65] |
| | | 1(c) | 2[64,66] |
| HIF2α P531A | No binding[a] | 1(c) | 1[57] |
| HIF2α L542P | Severe aggregation[b] | 1(b) | 1[63] |
| HIF2α D539E | Severe aggregation[b] | 2 | 1[67] |

[a]Aggregation observed at the highest peptide concentration, data point omitted, see Supplementary Fig. 3.
[b]Aggregation was too severe to measure the affinity, see Supplementary Fig. 3.

CODD T528M mutant, implying that there is some disruption of the interface relative to HIF1α due to the HIF2α T528 that can be restored by replacing T528 with a methionine. Interestingly, the HIF2α-CODD ΔG537 peptide had an identical affinity towards PHD2 as HIF1α-CODD WT (Fig. 2b and Table 2). These results suggest that while both sequence differences between HIF1α-CODD and HIF2α-CODD affect the interface with PHD2, the effects are not additive for determining the overall affinity of the interaction. This is particularly interesting when considering the location of Class 2 mutations, which are centered around G537, and that the residue G537 is one of the most commonly mutated residues in Class 2 disease (Table 2)[4]. Thus, given the importance of G537 in Pacak–Zhuang

syndrome and the differential affinity of PHD2 to HIF2α and HIF1α, which lacks a corresponding glycine residue, we decided to pursue a direct structural analysis of the PHD2/HIF2α-CODD interface.

## HIF1α-CODD and HIF2α-CODD peptides bind to similar PHD2 interface

We crystalized the PHD2/HIF2α-CODD complex in the presence of $FeSO_4$ and NOG to stabilize the interaction and prevent reaction turnover. We determined the crystal structure of the PHD2/HIF2α-CODD complex at a resolution of 1.8 Å (final $R_{work}$ = 17.3% and $R_{free}$ = 21.3%) via molecular replacement using a previous PHD2 structure[16] as a search model (Fig. 3a) (PDB code: 7UJV). The full summary of the data collection and refinement statistics can be found in Table 3. Well-defined electron density was observed for all 20 amino acids of the HIF2α-CODD peptide allowing for a detailed analysis of the binding interface (Fig. 3b). The HIF2α-CODD peptide is observed to fit into a cleft in the PHD2 surface with P531 positioned nearby to the coordinated $Fe^{2+}$ atom in the PHD2 active site and the PHD2 inhibitor NOG is observed to coordinate the $Fe^{2+}$ atom (Fig. 3c). The loop containing V241-K244 of PHD2 wraps around the top of the peptide, enclosing HIF2α P531 in the PHD2 active site (Fig. 3d). Overall HIF2α-CODD binds to PHD2 in a similar manner as HIF1α-CODD (PDB: 5L9B[16]) (Fig. 4a). PHD2 is nearly identical in both complex structures with a rmsd value of 0.6 Å for backbone atomic positions between the two structures (Fig. 4a). The main structural differences in PHD2 between the two structures are localized to the loop containing K244 and the end of the helix around Y403, which are both slightly repositioned relative to the PHD2/HIF1α-CODD complex structure to accommodate the HIF2α-CODD (Fig. 4a). Additionally, we observe a glycerol molecule from the cryoprotectant solution adjacent to the loop containing K244 which forms hydrogen bonds with the loop and the HIF2α-CODD peptide which may also influence the positioning of the loop. In agreement with our MST data, the main differences in the interface are surrounding HIF2α-CODD T528 and G537. In the PHD2/HIF1α-CODD complex, the HIF1α residue D558 resides within a short $3_{10}$-helical segment and forms a salt bridge with PHD2 K244 (Fig. 4b). In our HIF2α-CODD structure, the backbone carbonyl group of the corresponding D525 residue forms a hydrogen bond with the T528 side chain disrupting the helix structure and directing the D525 side chain away from PHD2 K244, not allowing the formation of a salt bridge (Fig. 4c). The presence of an additional glycine residue (i.e., G537) in the middle of the CODD sequence of HIF2α relative to HIF1α results in a reorganization in the surrounding residues that maintains the side chain interactions previously observed in the PHD2/HIF1α-CODD structure with some altered distances between the corresponding residues to accommodate the additional glycine residue (Fig. 4d and e). This adjustment of the positions of side chains results in an overall increase in the distance between the sulfur atom of HIF2α M535 and PHD2 Y403 by 0.6 Å relative to the corresponding HIF1α M568 and PHD2 Y403. This increased distance may explain the difference in the affinity of PHD2 for HIF1α-CODD and HIF2α-CODD as the M535-Y403 interaction would be weaker than the M568-Y403 interaction. Methionine-aromatic interactions have an optimal distance of ~5 Å; therefore, the M535-Y403 interaction is 0.9 Å greater than the optimal distance[19]. Thus, the presence of G537 in the HIF2α-CODD sequence indirectly affects the M535-Y403 interaction, reducing the affinity relative to HIF1α and indicates that the affinity of this interaction can be tuned by changes in this region of the peptide where we find most Class 2 mutations.

## Class 1 mutations disrupt critical interface residues between HIF2α and PHD2

We next sought to apply our new structural understanding of the PHD2/HIF2α-CODD interface to interpret our MST binding data. We observed that Class 1 mutations, which cause the most severe disruption to the PHD2/HIF2α-CODD interaction, are found in positions occupying or adjacent to the PHD2 active site (residues L529-Y532) and additionally D539 and L542. D539 forms ionic interactions with R396 and K399 of PHD2 (Fig. 4e),

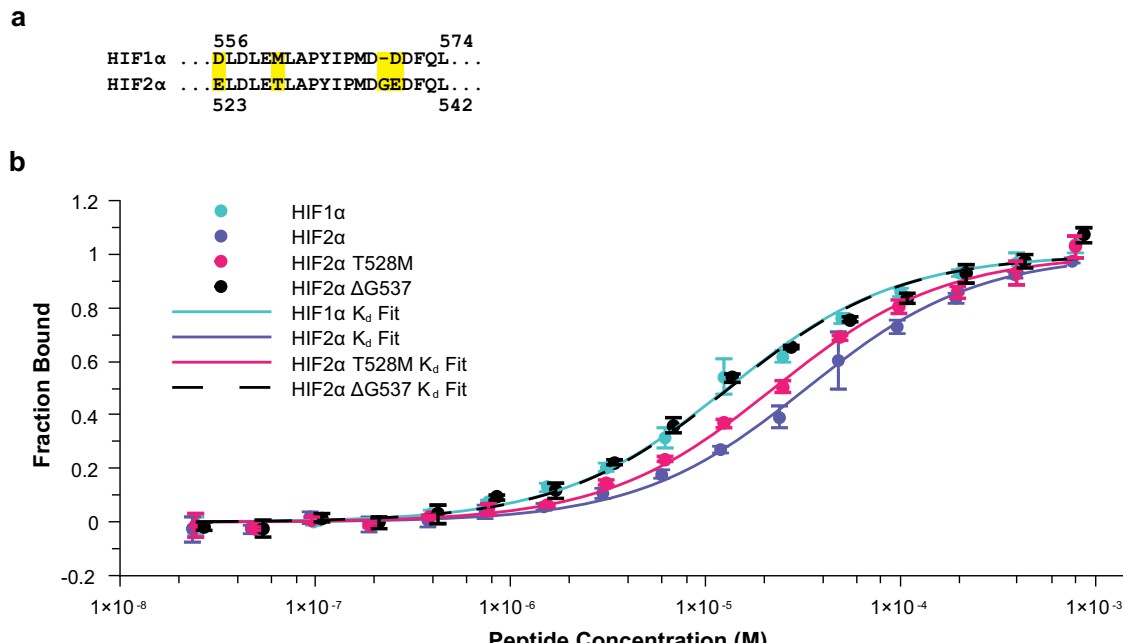

**Fig. 2 | Differences in the affinity of HIF1α-CODD and HIF2α-CODD for PHD2 are due to the insertion of a single glycine residue (G537) in the HIF2α-CODD sequence relative to HIF1α. a** Aligned sequences of the human HIF1α-CODD and HIF2α-CODD peptides with differences highlighted in yellow. **b** Binding curves of HIF1α and HIF2α-CODD peptides to PHD2 compared to binding curves of point mutants of HIF2α representing each independent difference between HIF1α and HIF2α. Individual non-normalized PHD2 MST binding curves for each peptide can be found in Supplementary Fig. 2.

mutations that disrupt these ionic interactions cause Class 1 disease (D539Y and D539N) while the mutation D539E causes Class 2 disease as this mutation likely maintains the ionic interactions. It should be noted that we are unable to comment on the effect of the D539E mutation on PHD2 affinity as the D539E peptide caused PHD2 to aggregate in MST experiments (Table 2 and Supplementary Fig. 3). L542 is located at the C-terminus of the HIF2α-CODD peptide and the side chain sits within a small hydrophobic pocket on the surface of PHD2. The peptide backbone of L542 forms a hydrogen bond with the PHD2 N293 carbonyl group (Fig. 5a). L542 has previously been shown to be required for HIF2α-CODD hydroxylation and these interactions are likely required to anchor the C-terminus of the CODD region to PHD2[4]. The L529P mutation examined would disrupt the short 3₁₀-helix as it would remove the only helix backbone hydrogen bond between L529 and L526 as well as a hydrogen bond between the L529 carbonyl oxygen and the side chain of PHD2 Y310 (Fig. 5b and c). In an unbound PHD2 structure, the loop containing V241-K244 is in an alternative confirmation which leaves the PHD2 active site accessible to the solvent[16]. Thus, a conformational change occurs upon binding as this loop is observed to wrap around HIFα-CODD in our PHD2/HIF2α-CODD complex structure as well as previous PHD2/HIF1α-CODD structures (Figs. 3d and 4a)[15,16]. Mutations to residue A530, which is separated from the solvent by this loop in our structure, would inhibit this reorganization, especially mutations to bulkier residues. This is indeed what is observed as while both A530V and A530T have a significant effect on PHD2 binding affinity, A530V is more detrimental to the interaction than A530T (Fig. 1c and Table 2). Disruption of this conformational change could also disrupt hydrogen bonds between HIF2α-CODD A530 and PHD2 V241 and HIF2α-CODD Y532 and PHD2 Q239 (Fig. 5c). Mutations of residue P531 eliminate the possibility of hydroxylation as the target proline residue is substituted from the sequence, however we additionally observed that all P531 mutants examined in the present study do not bind to PHD2. This lack of interaction is likely due to the increased flexibility in the peptide backbone that other amino acids would impart relative to a proline residue at this position, which may potentially disrupt the interaction between the PHD2 R322 sidechain and the substituted residues' backbone carbonyl group

(Fig. 5c). Mutations at Y532, which is adjacent to the active site (Fig. 5d), can cause either Class 1 or Class 2 disease depending on the mutation. Y532C causes Class 1 disease while Y532H causes Class 2 disease. In agreement with our binding data, only Y532C is strongly detrimental to the interaction of PHD2 with HIF2α-CODD (Fig. 2c and Table 2). The reason for this amino acid-specific effect is not immediately clear as most of the Y532 side chain does not make direct contact with PHD2 and instead faces the solvent (Fig. 5d). A likely explanation is that the Y532 side chain excludes solvent from the active site, stabilizing the interaction and that mutations to similarly bulky amino acids such as histidine maintain the exclusion of solvent, unlike smaller amino acids such as cysteine. Thus, the structure of the PHD2/HIF2α-CODD complex provides clear mechanisms by which mutations to these critical residues disrupt the interaction between PHD2 and HIF2α-CODD.

## Most Class 2 mutations disrupt the more flexible region of HIF2α-CODD

Class 2 mutations are found in the middle region of HIF2α-CODD peptide at positions Y532-F540 except for positions D536 and E538 for which there are no known disease-causing mutations. We observed that this region of the peptide does not make as much contact with the surface of PHD2 as residues that cause Class 1 disease (Fig. 5d). There is a sharp increase in atomic B-factor values in residues P534-D539 relative to the rest of the peptide indicating that this segment of the peptide has increased flexibility, likely due to the reduced contact with PHD2 (Fig. 5e). Most Class 2 disease mutations occur on residues M535 and G537[4] and other Class 2 mutations are centered around residue G537 (Table 2), localizing a large proportion of these mutations to this middle flexible segment of the peptide (Fig. 5d and e). Mutations in this region of the peptide would have a lesser effect on PHD2 binding as it is more loosely associated with PHD2 than the flanking sequence, which is consistent with our MST binding data (Fig. 2c). We further compared normalized B-factors (modified Z-score) of HIF2α-CODD from the present study to the previous PHD2/HIF1α-CODD structure[16] to ascertain if this region is more flexible than the corresponding region of the HIF1α-CODD. There is an observed increase in the flexibility

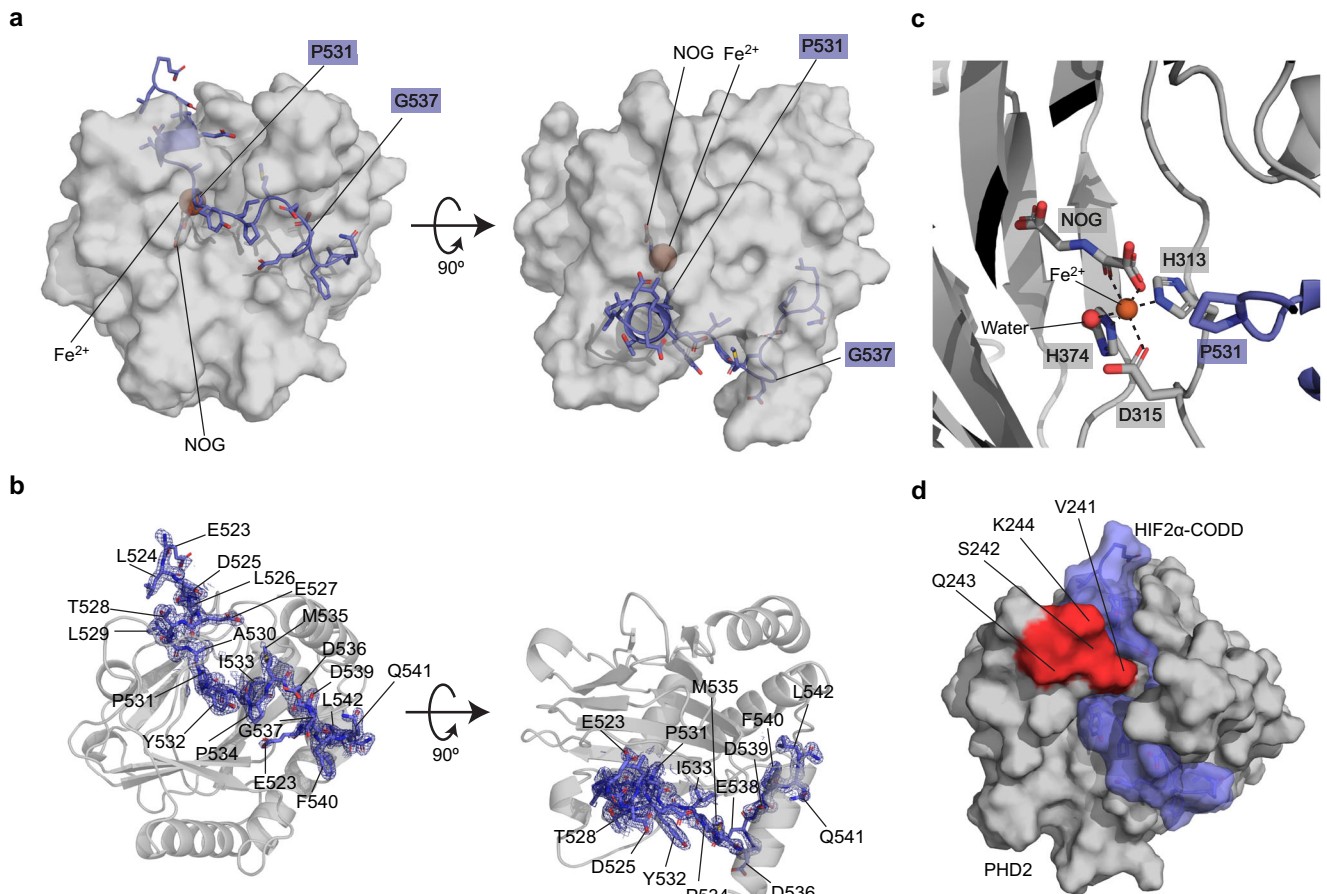

**Fig. 3 | Crystal structure of the PHD2/HIF2α-CODD complex to 1.8 Å resolution. a** Overview of the structure of the PHD2/HIF2α-CODD complex. PHD2 is depicted by gray transparent surface and HIF2α-CODD is depicted as a slate-colored ribbon representation with side chains shown as sticks. Small molecules and key residues are indicated. **b** Composite omit electron density of the HIF2α-CODD peptide. The co-crystal structure of the PHD2/HIF2α-CODD complex is shown with PHD2 represented by a transparent gray cartoon model and the HIF2α-CODD peptide as slate-colored stick model. The composite omit electron density map $(2F_O-F_C)$ is contoured at 1.0 σ and is shown as a blue mesh surrounding the peptide. **c** The PHD2 active site depicting the coordinated $Fe^{2+}$ atom, NOG and HIF2α P531. **d** Surface representation of PHD2 (Gray) and HIF2α-CODD (transparent slate). The location of PHD2 residues V241-K244 on the surface model is indicated (red) displaying how the loop wraps around HIF2α-CODD.

of the residues M535-E528 as compared to the corresponding HIF1α-CODD residues M568-D570 (Fig. 5f). Of these HIF2α residues with higher flexibility, only M535 and E538 are positioned in our model in such a way as to suggest they may form stabilizing interactions with PHD2 (Fig. 4e). However, the atomic B-factor values indicate that E538 is one of the most flexible residues in the peptide. This is supported by the relatively poor electron density observed for the E538 side chain (Fig. 3b). This suggests that this interaction is weaker than would be expected for a stable salt bridge between HIF2α E538 and PHD2 K297 and may explain why no known disease-associated mutations occur at this position (Fig. 5d). The increased flexibility of HIF2α M535 relative to the corresponding HIF1α M568 residue agrees with the above discussed weaker M535-Y403 interaction in our PHD2/HIF2α-CODD complex. Thus, disruption of the M535-Y403 interaction would eliminate the main stabilizing interaction in this already flexible region of the HIF2α-CODD and is likely the cause of the reduced affinity of Class 2 mutations occurring at M535. Furthermore, Class 2 mutations found at the preceding two residues, I533 and P534, likely disrupt the correct positioning of M535 in proximity to Y403 of PHD2. I533 occupies a pocket on the PHD2 surface (Fig. 5g), which likely stabilizes the overall interaction and correctly positions the peptide backbone. P534 provides structural rigidity for the observed bend of the peptide backbone necessary to position HIF2α M535 beneath PHD2 Y403.

Missense mutations at G537 would result in the replacement of the G537 with bulkier and less flexible residues. The turn in the peptide

backbone at G537 requires the adoption of torsional angles in the left-handed helix region of the Ramachandran plot (Supplementary Fig. 4), which has been found to be largely populated by glycine residues in large-scale investigations of torsional angles in protein structures[20]. Importantly, while the torsional angles of G537 are within the allowed region of the Ramachandran plot for most amino acids, it is extremely rare for tryptophan and arginine residues to adopt similar torsional angles in known loop structures[20]. Therefore, the G537W and G537R mutations would favor the adoption of alternate conformations and disrupt the interactions of the nearby residues. From our knowledge of HIF2-disease phenotypes we propose that these mutations would further disrupt the M535-Y403 interaction. This would result in comparable $K_d$ values for mutations to residues G537, I533, P534, and M535 as the mechanism of disruption would be similar; and concordantly, we observed an overall narrow range of $K_d$ values for most Class 2 mutants (Fig. 1c and Table 2). The remaining two positions of Class 2 mutations occur at residues D539 and F540. As mentioned above, we were unable to measure the affinity of D539E to PHD2 due to observed aggregation and cannot comment on the effect of this mutation on the interface. We observe that mutations at F540 to non-aromatic residues such as F540L would disrupt the π-cation interaction between HIF2α F540 and PHD2 R295 (Fig. 5a). The atomic B-factor values decrease in this region (Fig. 5e) implying that F540 along with D539 and L542 stabilize the PHD2/HIF2α-CODD interaction; however, this interpretation may be complicated by a nearby stabilizing

**Table 3 | Data collection and refinement statistics for PHD2/HIF2α-CODD crystal structure**

| | PHD2/HIF2α-CODD (7UJV) |
|---|---|
| *Data collection* | |
| Space group | P2₁2₁2 |
| *Cell dimensions* | |
| a, b, c (Å) | 129.08, 37.60, 42.15 |
| α, β, γ (°) | 90, 90, 90 |
| Resolution (Å) | 42.18–1.80 (1.83–1.80)[a] |
| $R_{meas}$ (%)[b] | 12.6 (86.0)[a] |
| $R_{pim}$ (%)[c] | 4.7 (33.4)[a] |
| I/σ(I) | 8.4 (0.8)[a] |
| CC₁/₂ (%) | 99.8 (69.5)[a] |
| Completeness (%) | 99.8 (99.2)[a] |
| Total no. of reflections | 140391 (6308)[a] |
| No. of unique reflections | 19834 (967)[a] |
| Redundancy | 7.1 (6.5)[a] |
| *Refinement* | |
| Resolution (Å) | 42.15–1.80 |
| $R_{work}$/$R_{free}$ (%)[d,e] | 17.3/21.4 |
| No. of atoms (non-hydrogen) | 2014 |
| Protein | 1888 |
| Ligand/Ion | 29 |
| Water | 108 |
| B-factor (Å²) | 29.3 |
| Protein (Å²) | 28.9 |
| Ligand/Ion (Å²) | 28.4 |
| Water (Å²) | 38.0 |
| *R.m.s. deviation* | |
| Bond lengths (Å) | 0.009 |
| Bond angles (°) | 1.49 |
| *Ramachandran plot* | |
| Most favored region (%) | 97.4 |
| Additional allowed region (%) | 2.6 |
| Outliers (%) | 0.0 |
| Rotamer outliers (%) | 3.48 |
| Clashscore | 3.2 |

[a]Values in parentheses represent statistics for the highest resolution shell.

$$^{b}R_{meas} = \frac{\sum_{hkl} \sqrt{\frac{n}{n-1}} \sum_{i=1}^{n} |I_{hkl,i} - \bar{I}_{hkl}|}{\sum_{hkl} \sum_{i=1}^{n} I_{hkl,i}}$$

$$^{c}R_{pim} = \frac{\sum_{hkl} \sqrt{\frac{1}{n-1}} \sum_{i=1}^{n} |I_{hkl,i} - \bar{I}_{hkl}|}{\sum_{hkl} \sum_{i=1}^{n} I_{hkl,i}}$$

$$^{d}R = \frac{\sum |F_{obs} - F_{calc}|}{\sum F_{obs}}$$

[e]$R_{free}$ is calculated using a separated test set of reflections omitted from model refinement (4.79% of all reflections) and $R_{work}$ is calculated using all reflections used in model refinement.

hydrogen-bond between the amidated C-terminus of the peptide and the neighboring PHD2 molecule in the crystal lattice.

### Class 1 mutations increase HIF2α transcriptional activity more than Class 2 mutations

We next asked whether the extent to which disease-causing mutations diminish HIF2α binding to PHD2 proportionally increases the transcriptional activity of HIF2 using dual luciferase reporter under the control of hypoxia-responsive element (HRE). We showed that ectopic Class

1-associated HIF2α mutants induced stronger HRE-driven transcriptional activity under normal oxygenated conditions compared to Class 2 or WT HIF2α (Fig. 6a). The transcriptional activity of Class 2 mutants was statistically indistinguishable from WT HIF2α (Fig. 6a). These results suggest that the more severe defect in HIF2α binding to PHD2 due to Class 1 mutations compared to Class 2 mutations, which would lead to less prolyl-hydroxylation and thus enhanced stabilization of HIF2α, correlates to increased HIF2α-dependent transcription. Moreover, HIF2αP405A/P531A double mutant displayed indistinguishable transcriptional activity from HIF2αP531A single mutant under normoxic conditions while the P405A mutant displayed transcriptional activity comparable to wild-type HIF2α suggesting that the elimination of P405 hydroxylation within the N-terminal ODD site offers negligible activation of HIF2α under these conditions (Fig. 6b).

## Discussion

Pacak-Zhuang syndrome is a pseudohypoxic disease resulting from the inappropriate stabilization of HIF2α under normoxic conditions caused by gain-of-function missense mutations in the CODD region of HIF2α. These missense mutations stabilize HIF2α by ultimately inhibiting proteasome-mediated destruction of HIF2α[2,6,21]. Previously, we hypothesized that the mutations in the HIF2α-CODD region disrupted the pVHL/HIF2α interface. This appeared to be true for most Class 1 mutants examined to date. However, most Class 2 mutant peptides displayed a binding affinity for pVHL that was similar or indistinguishable from WT HIF2α-CODD peptide[4]. Notably, we observed that the Class 1 mutations A530V and L542P seemed to have no effect on pVHL binding but instead appeared to decrease PHD-mediated hydroxylation[4]. There are three PHD paralogs in humans (PHD1-3) with PHD2 being the most abundantly expressed in many cell types[22]. Furthermore, numerous (>95) germline PHD2 mutations have been reported to be causative of erythrocytosis with or without PPGL suggesting that PHD2 is central to HIF2α hydroxylation in health and disease[5,23–25]. PHD1 and PHD3 do not appear to play an entirely redundant role with PHD2 as no known erythrocytosis- or PPGL-associated mutations have been identified in PHD3 and only 1 case has been reported with a PHD1 mutation[25]. Thus, the overall impact of PHD-mediated hydroxylation of HIF2α in Pacak-Zhuang syndrome required further investigation with a focus on PHD2, the major enzyme mediating HIF2α hydroxylation.

Here, we used MST to determine the $K_d$ of the interaction between PHD2 and HIF2α-CODD peptides in solution. We observed that all HIF2α mutants examined had higher $K_d$ values than WT HIF2α-CODD peptide (Table 2). Class 2 mutants had $K_d$ values 1.2–6.5× the WT value with most values within a narrow range between 4.4–6.5× WT (Fig. 1c and Table 2). Class 2 mutations Y532H and G537W had a milder effect on PHD2/HIF2α-CODD $K_d$ with values at 1.2× and 2.3× WT, respectively (Table 2). Notably, we found that Class 1 mutants had $K_d$ values > 10× the WT value or did not bind to PHD2. These results demonstrate a clear quantifiable distinction between WT, Class 1 and Class 2 mutant HIF2α peptides (Table 2). Thus, while it is probable that the disease manifestations of Pacak–Zhuang syndrome arise because of the reduced affinity of PHD2 for the mutant HIF2α in addition to the disruption of pVHL-mediated binding and poly-ubiquitylation of hydroxylated mutant HIF2α, the impact of these mutations on the affinity of PHD2 for HIF2α appears to be sufficient to differentiate between Class 1 disease- and Class 2 disease-causing mutations.

Based on our PHD2/HIF2α-CODD co-crystal structure, which is similar to the other recent structures[26], Class 1 mutations affect residues that are adjacent to the PHD2 active site, D539 and L542 (Fig. 5d). It can be appreciated that mutations to these residues would be very disruptive to the interface with PHD2 as they are observed to either participate in stabilizing interactions and/or occupy restrictive spaces that cannot accommodate alternative residues due to predicted steric clashes (Fig. 5a–c). Additionally, some Class 1 mutations occur at P531 eliminating the possibility of hydroxylation and therefore any binding to pVHL. Class 2 mutations affect residues that are generally more flexible and make less contact with PHD2 (Fig. 5d and e). From our analysis we were able to identify HIF2α M535 as a

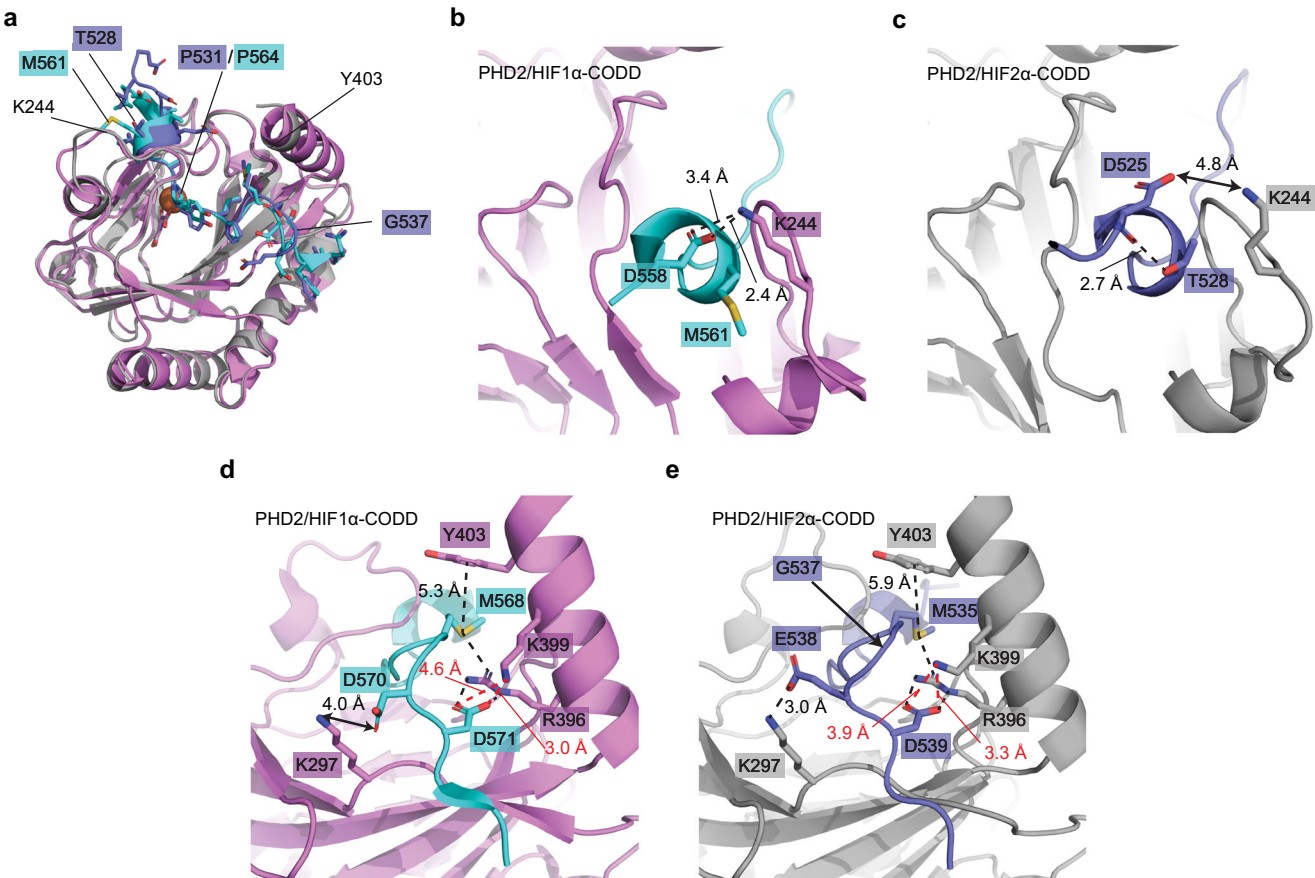

**Fig. 4 | HIF1α-CODD and HIF2α-CODD peptides bind to similar PHD2 interface with key differences. a** Aligned structures of PHD2/HIF1α-CODD complex (PDB: 5L9B) with the structure of the PHD2/HIF2α-CODD complex. PHD2 from our HIF2α-CODD complex model is depicted as gray ribbon model, while PHD2 from the HIF1α-CODD complex is colored in violet. HIF1α-CODD (cyan) and HIF2α-CODD (slate) are depicted as cartoon ribbon models with side chains shown as sticks. **b** The N-terminal region of the PHD2/HIF1α-CODD interface highlighting the salt bridge formed by D558 on HIF1α and K244 on PHD2. HIF1α M561 which aligns to the same position as HIF2α T528 is also displayed. **c** The N-terminal region of the PHD2/HIF2α-CODD interface highlighting the

interaction between the HIF2α T528 side chain and the carbonyl group of the HIF2α D525 residue which aligns to HIF1α D558. This positions D525 in such a way as to disallow a salt bridge to form between HIF2α D525 and PHD2 K244. **d** Middle region of the HIF1α-CODD PHD2 interface with side chain interactions depicted as either black or red dashed lines. **e** Middle region of the HIF2α-CODD PHD2 interface with side chain interactions highlighted as either black or red dashed lines. In panels **d** and **e** only interaction distances that differ between the two structures are labeled. Measured distances that do not describe interactions are depicted as solid lines with arrows on each end.

key residue in modulating the affinity of HIF2α-CODD for PHD2 due to its stabilizing interactions with PHD2 R396 and Y403 (Fig. 4e). Most Class 2 mutations either directly disrupt this interaction by altering the methionine residue to another amino acid or are predicted to indirectly interfere with the correct positioning of M535 in proximity to PHD2 Y403 and R396. These binding data in concert with structural insight from PHD2/HIF2α-CODD co-crystal structure provide an updated understanding of the mechanisms underlying HIF2α stabilization in Pacak-Zhuang syndrome.

The observed differential effect of Class 1 and Class 2 mutations on the transcriptional activation by HIF2α supports the biophysical assessment that Class 1 mutations are more disruptive to the oxygen-mediated destruction of HIF2α and lead to increased HIF2α transcriptional activity (Fig. 6a). Importantly, the transcriptional activity of disease-causing HIF2α mutations was observed to correlate to the degree to which the mutation diminished the affinity to PHD2 (Table 2, Fig. 1c and Fig. 6a). The lack of any disease-causing mutations surrounding P405 suggests that the N-terminal hydroxylation site is not critical for the proper regulation of HIF2α stability and transcriptional activity. We examined the hydroxylation-deficient mutants P405A, P531A, and P405A/P531A relative to wild-type HIF2α and observed that there was no significant increase in HIF2α transcriptional activation activity caused by P405A mutation relative to wild-type HIF2α. Furthermore, the single proline mutant P531A showed

a similar increase in transcriptional activity as the double mutant HIF2αP405A/P531A (Fig. 6b). These results suggest that P405 hydroxylation has a minimal impact on the regulation of HIF2 transcriptional activity on target genes. A similar result was observed in a previous study of the HIF1α N-terminal target proline suggesting that the C-terminal proline residue is the critical regulatory component for oxygen-mediated destruction in both HIF1α and HIF2α paralogs[27]. Moreover, we did not observe a statistically significant increase in HRE-dependent transcriptional activation by Class 2 HIF2α mutants compared to WT HIF2α despite Class 2 HIF2α mutants having a measurable, albeit subtle, defect in binding PHD2 (Fig. 6a).

We propose that the disruptions to the PHD2/HIF2α-CODD interface and therefore prolyl-hydroxylation via PHD2 is arguably the most influential step in the observed HIF2α stabilization in Pacak-Zhuang syndrome. PHD-mediated hydroxylation is, however, not the only disrupted process as evidenced by the near wild-type affinity of PHD2 for the HIF2αY532H mutant and the previously described disruptions to pVHL recognition of mutant HIF2α-CODD peptides[4]. The effect of HIF2α mutations on the $K_d$ of the PHD2/HIF2α-CODD interaction is clearly segregated by the general disease class in Pacak-Zhuang syndrome. Thus, we argue that for mutations in the HIF2α-CODD region, the broad disease class can be determined by the $K_d$ of the interaction of the mutant HIF2α-CODD with PHD2.

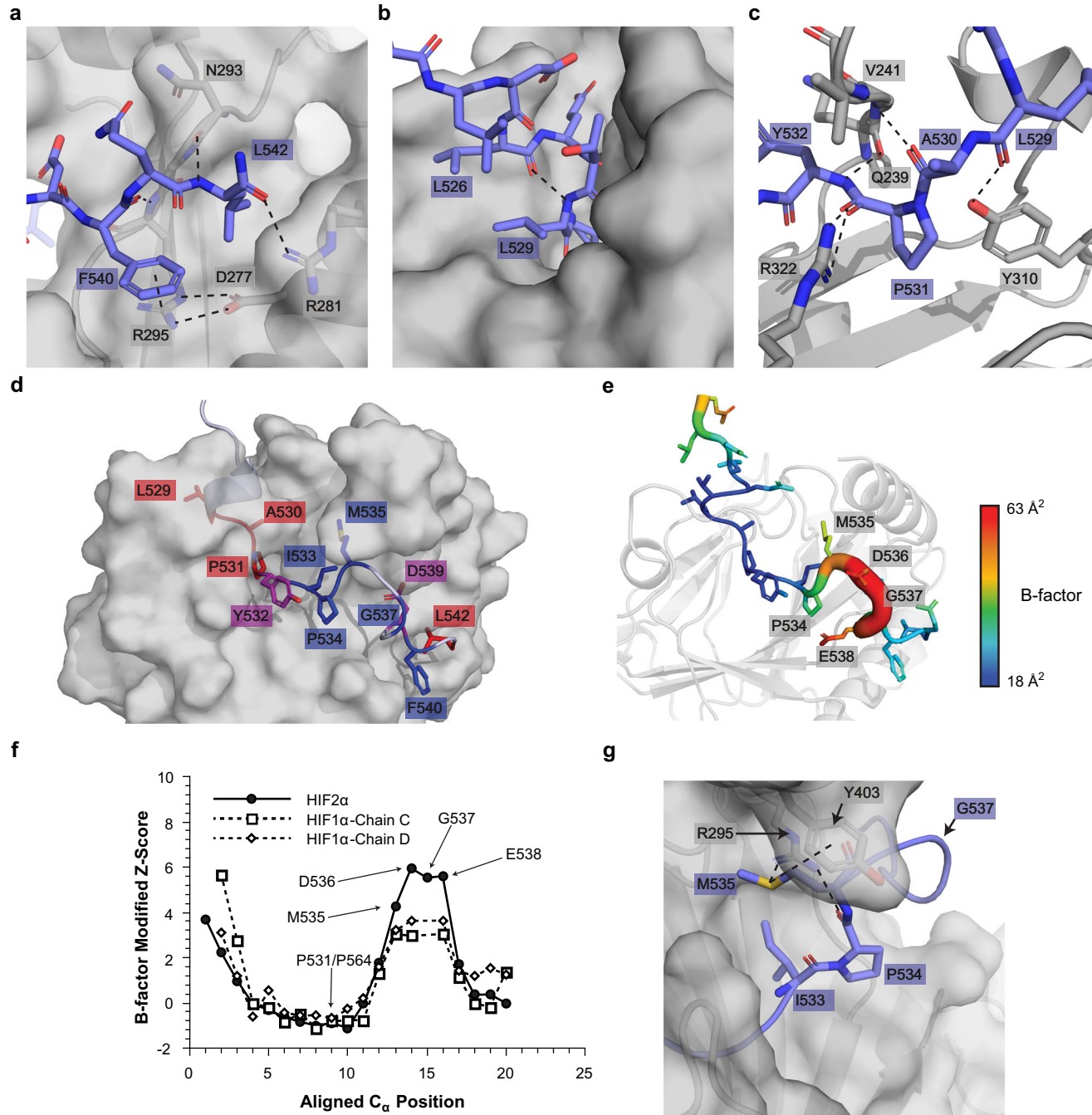

**Fig. 5 | Class 1 mutations disrupt key interactions between HIF2α and PHD2 while Class 2 mutations are localized to the flexible middle region of HIF2α-CODD peptide. a** Interactions at the C-terminus of the HIF2α-CODD peptide, the L542 side chain occupies a small pocket on the PHD2 surface. **b** The stabilizing hydrogen bond between L526 and L529 of HIF2α-CODD. **c** The hydrogen bonding network adjacent to HIF2α-CODD P531 and the PHD2 active site. **d** PHD2/HIF2α-CODD complex structure with the residues mutated in Class 1 (red), Class 2 (blue) and both classes (purple) of Pacak–Zhuang syndrome. **e** Atomic *B*-factor representation of the HIF2α-CODD peptide from the PHD2/HIF2α-CODD co-crystal structure. HIF2α-CODD is colored by the atomic *B*-factor and the backbone diameter increases with $C_\alpha$ *B*-factor. PHD2 is depicted as a transparent gray ribbon.

Residues with relatively high B-factors, which indicate flexibility in the structure, are labeled and colored red. **f** Normalized $C_\alpha$ B-factors (modified *Z*-score) of the aligned HIF2α-CODD and HIF1α-CODD (PDB Code: 5L9B)[16]. Both copies of the HIF1α-CODD in the asymmetric unit of the PHD2/HIF1α-CODD structure are included and are labeled by their PDB chain IDs (Chain C and Chain D). Residues of HIF2α-CODD observed to have elevated normalized $C_\alpha$ B-factors relative to HIF1α-CODD are labeled in addition to P564 and P531 of HIF1α and HIF2α, respectively. **g** HIF2α-CODD residues adjacent to M535 involved in Class 2 Pacak–Zhuang syndrome. The molecular surface of PHD2 (gray) is superimposed with a ribbon diagram. The Y403 side chain is shown in stick representation. In panels **a–c** and **g**, dashed lines represent hydrogen bonds, salt bridges or π-cation interactions.

Furthermore, de novo detrimental mutations can be assessed using our PHD2/HIF2α-CODD structure to determine or predict how they may disrupt PHD2/HIF2α-CODD interactions. This could be of importance in the clinical surveillance and management of patients presenting with

sporadic instances of polycythemia linked to novel HIF2α mutations as the further risk of PPGL and somatostatinoma, which occur decades later than the onset of polycythemia in Class 1 disease[4], could be theoretically predicted.

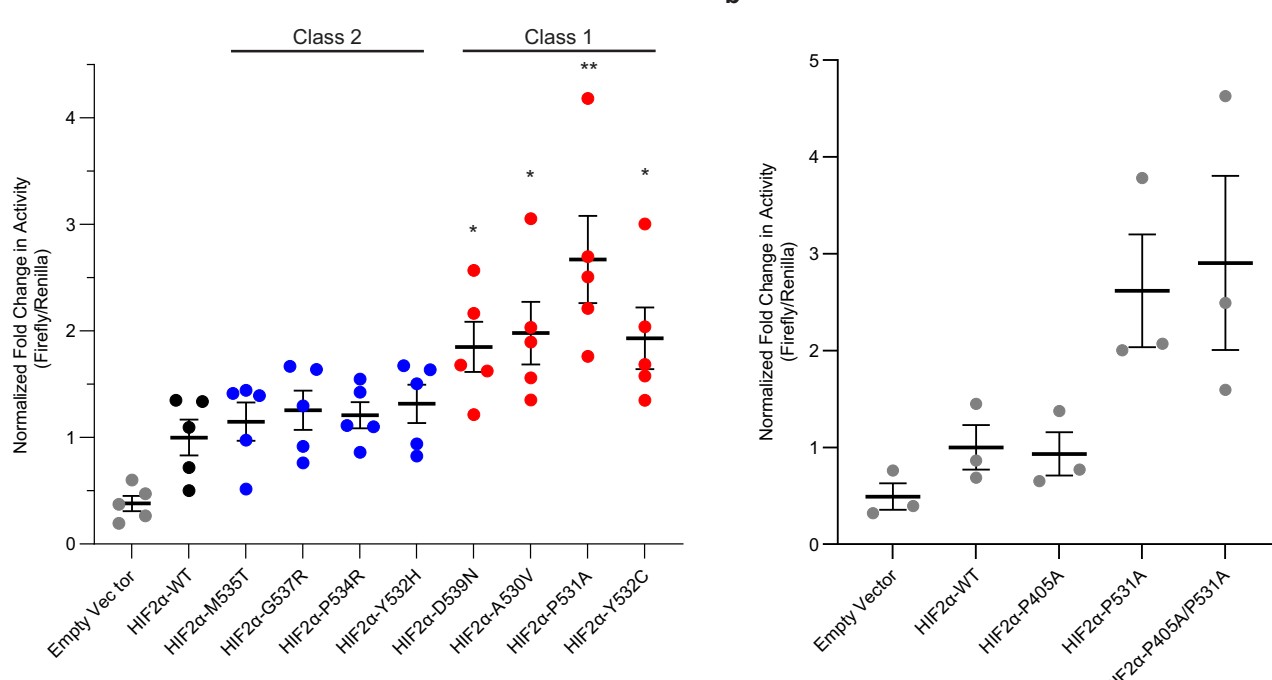

**Fig. 6 | Pacak–Zhuang syndrome Class 1 mutations impart elevated HIF2α transcriptional activity in HEK293A cells. a** Dual luciferase reporter measurements of Class 1 and Class 2 mutant HIF2α transcriptional activity relative to HIF2α WT. * Indicates $P < 0.05$ and ** indicates $P < 0.01$ (two-tailed $t$-test). **b** Dual luciferase reporter measurements of HIF2α P531A (CODD site mutant), HIF2α P405A (NODD site mutant) and HIF2α P405A/P531A (NODD and CODD site double mutant) transcriptional activity relative to HIF2α WT. In both panels **a** and **b**, individual data points are plotted, horizontal lines represent mean values, and error bars represent standard error.

## Methods
### Plasmids, antibodies, and peptides

The following previously described plasmids were used in this study: pcDNA3-HA-VHL₃₀[28] and pET-46-HIS₆-PHD2[4]. pGL3-VEGFA-HRE was generated by amplifying nucleotides 1271-1797 of human VEGFA CDS (AH001553.2) using primers introducing 5' KpnI and 3' XhoI sites; digested PCR products were subcloned into pGL3-Basic (Promega) cut with KpnI and XhoI. pRL-SV40 was obtained from Promega. pcDNA3-HA-HIF2α has been previously described[28]. Class 1 (D539N, A530V, P531A, Y532C) and Class 2 (M535T, G537R, P534R, Y532H) HIF2α mutants, as well as mutant P405A, were generated using QuikChange site-directed mutagenesis (Agilent) and confirmed by DNA sequencing. pcDNA3-HA-HIF2α-P405A/P531A was obtained from Addgene (Addgene plasmid # 18956)[29]. The α-HA tag antibody was obtained from Cell Signaling (C29F4). HIFα-CODD peptides were custom synthesized by Genscript and supplied at ≥ 95% purity. All synthesized peptides were C-terminally amidated, peptides used for MST binding studies and crystallography were N-terminally acetylated and peptides used for the indirect hydroxylation assays were N-terminally biotinylated. For crystallography and MST experiments peptides were dissolved directly into 50 mM Tris pH 8.0 buffer and for indirect hydroxylation assays N-terminally biotinylated peptides were initially dissolved in DMSO to create stock solutions and further diluted into aqueous buffers for immobilization on streptavidin agarose beads. The sequences of each peptide can be found in Supplementary Table 1.

### Purification of PHD2

PHD2 was purified via a previously described but modified procedure[4,30]. *E. coli* BL21(DE3) cells were transformed with pET-46-HIS₆-PHD2 which encodes for His₆-PHD2 (prolyl-hydroxylase domain, residues 181–426). Cells were grown to an OD₆₀₀ of 0.8 at 37 °C and expression of His₆-PHD2 was induced with a final concentration of 0.5 mM IPTG. Cells were grown for an additional 3 h post-induction at 37 °C and harvested via centrifugation. Cell pellets were either used immediately for purification or stored at −80 °C for later use. Cell pellets were resuspended in Lysis Buffer (50 mM Tris–HCl pH 7.9, 500 mM NaCl, 5 mM Imidazole) containing 1x SigmaFast protease inhibitor cocktail (Sigma-Aldrich) and lysed with 3 passes through an Emulsiflex-C3 cell disruptor (Avestin) at 20-30 kPSI. Cell lysates were clarified via centrifugation at 16,000 rpm in a JA-25.50 rotor (Beckman Coulter). Clarified lysates were applied to a 2 ml Ni-NTA (Thermo Scientific) column pre-equilibrated with Lysis Buffer. The beads were washed with 10 ml Lysis buffer and 2 × 30 ml of Wash Buffer (50 mM Tris–HCl pH 7.9, 500 mM NaCl, 30 mM Imidazole). His₆-PHD2 was eluted by applying 10 ml Elution Buffer (50 mM Tris–HCl pH 7.9, 500 mM NaCl, 1 M Imidazole) to the column. Ni-NTA purified His₆-PHD2 was dialyzed using a 10 kDa cut-off dialysis membrane against 2 L of Dialysis Buffer (50 mM Tris–HCl pH 7.5) for 4 h at 4 °C. After 4 h, the concentration of His₆-PHD2 was measured by UV absorbance at 280 nm ($\varepsilon_{0.1\%} = 1.34$), 1-1.5 units of thrombin were added to the dialysis bag per mg of His₆-PHD2, transferred to fresh 2 L Dialysis Buffer. Dialysis and thrombin cleavage were allowed to continue overnight at 4 °C. Thrombin-cleaved PHD2 was collected and applied to a 2 ml Ni-NTA column pre-equilibrated with Dialysis Buffer to remove uncleaved protein and cleaved His₆ tags. The flow-through was collected and the beads were washed with 10 ml of Dialysis Buffer supplemented with 5 mM imidazole. The 5 mM imidazole eluate was pooled with the flow through and concentrated with an Amicon 10 kDa cut-off centrifugal concentrator (Millipore) to a volume of ~600 μl. The concentrated PHD2 was then applied to a Superdex 75 Increase 10/300 GL column (Cytiva) equilibrated with Dialysis Buffer or Labeling Buffer (50 mM HEPES pH 7.5, 150 mM NaCl) depending on the intended use of the sample. Eluted fractions containing PHD2 were collected and pooled. For the His₆-PHD2 used in the indirect hydroxylation assay, no thrombin was added during dialysis steps therefore the His₆-tag was not removed and the subsequent passage of the sample through Ni-NTA beads to remove His₆ tags was not performed. Otherwise, the purification protocol for His₆-PHD2 is the same as thrombin-cleaved PHD2. All purification steps were analyzed by SDS–PAGE and the Superdex 75 Increase chromatogram was

used to assess size and aggregation, see Supplementary Fig. 5 for a representative purification assessment.

## Indirect hydroxylation assay

The indirect hydroxylation assay was performed as reported previously[4,31]. Briefly, 4 μg of biotinylated HIFα-CODD peptide was immobilized on streptavidin-agarose beads. Following immobilization, the beads were washed 2× with EBC buffer (50 mM Tris–HCl, 120 mM NaCl, 0.5% (v/v) NP-40) and then 2× with 50 mM Tris–HCl pH 7.5. The immobilized peptides were then incubated with 7.5 μg HIS$_6$-PHD2 (residues 181–426) in 500 μl of 50 mM Tris–HCl pH 7.5, 300 μM α-ketoglutarate, 2 mM ascorbic acid, 1 mM dithiothreitol, and 150 μM FeCl$_2$ at room temperature for 1.5 h with gentle agitation. The beads were washed 5× with EBC buffer. Immobilized peptides were then incubated with HA-pVHL$_{30}$, produced via in vitro transcription and translation in rabbit reticulocyte lysate (Promega), at 4 °C for 1.5 h in EBC buffer with 1× EDTA-free SigmaFast protease inhibitor (Sigma-Aldrich). The beads were again washed 5× with EBC buffer. Immobilized material was boiled in 1× SDS-PAGE sample buffer to elute protein and HA-pVHL$_{30}$ protein amounts were examined via Western blot as a surrogate for hydroxylation.

## Alexa-647 labeling of PHD2

A 500 μl aliquot of 1 mg/ml PHD2 in Labeling Buffer was labeled with Alexa Fluor 647 N-hydroxysuccinimide ester dye (Thermo Scientific) by addition of a 4:1 molar ratio of dye:PHD2. Labeling was conducted at room temperature for 1 h in the dark. Excess dye was then removed first by the passage of the sample through a PD-10 desalting column containing Sephadex G-25 resin (Cytiva) equilibrated with Dialysis Buffer and then any remaining excess dye was removed via repeated 10-fold dilution in Dialysis Buffer and subsequent concentration in an Amicon 10 kDa cut off centrifugal filter until the dye could no longer be detected in the concentrator flow through by Absorbance at 650 nm. This procedure should result in Alexa 647 labeled PHD2 (PHD2-647) with a labeling efficiency of 1-2 dye molecules per protein molecule. PHD2-647 was diluted to a concentration of 10 μM in Dialysis Buffer, aliquots were used immediately, or flash frozen in liquid N$_2$ and stored at −80 °C until needed.

## Microscale thermophoresis

MST measurements were taken using a Monolith NT.115 (Nanotemper Technologies) at room temperature using standard MST capillaries (Nanotemper Technologies). A 2x stock of PHD2-647 was created by diluting PHD2-647 from a 10 μM stock to 50 nM with MST Buffer (50 mM Tris–HCl pH 7.5, 1 mM NOG, 10 μM FeSO$_4$, 5 mg/ml BSA). For each HIFα-CODD peptide, ~1 mg dry weight of peptide was dissolved in 200 μl of 50 mM Tris–HCl pH 8.0 and concentrations were measured using UV absorbance at 280 nm for most peptides and BCA assay (Thermo Scientific) for Y532C and Y532H peptides using known concentrations of the WT HIF2α-CODD peptide to create a standard curve. The UV absorbance extinction coefficients for all HIF1α-CODD and HIF2α-CODD peptides used in this study is 1490 M$^{-1}$ cm$^{-1}$ except for HIF2α-CODD D539Y (2980 M$^{-1}$ cm$^{-1}$), HIF2α-CODD G537W (6990 M$^{-1}$ cm$^{-1}$) and HIF2α-CODD Y532C and Y532H which were not analyzed via absorbance at 280 nm due to the lack of any Tyrosine or Tryptophan residues in the peptides. For each experiment, 16 different concentrations of 15 μl peptide solution were created by 2-fold serial dilution of the peptide stock in 50 mM Tris–HCl pH 8.0. Each of the 16 peptide solutions was then mixed with an equal volume (15 μl) of 2× PHD2-647 solution resulting in a final 1× concentration of PHD2-647 at 25 nM. Solutions were incubated at room temperature for 10 min and then loaded into MST capillaries for measurement. MST measurements were taken at 20% fluorescence excitation LED power and 40% (medium) infrared laser power. For each MST time trace initial fluorescence (pre-infrared laser activation) was recorded for 5 s, infrared laser on time was 20 s, and fluorescence post-infrared laser deactivation was recorded for 3 s. MST data was analyzed using the M.O. Affinity Analysis software (Nanotemper Technologies) and the 1:1 $K_d$ binding

model was fit to experimental data. The 1:1 $K_d$ binding model as it pertains to MST has been previously described in detail[13,14,32], a summary is as follows:

$$F_{\mathrm{n}} = 1000 \left( \frac{F_{\mathrm{hot}}}{F_{\mathrm{cold}}} \right) \tag{1}$$

$$F_{\mathrm{n}}(A) = F_{\mathrm{B}} + \frac{\left( F_{\mathrm{AB}} - F_{\mathrm{B}} \right) \left( A + B + K_{\mathrm{d}} - \sqrt{\left( A + B + K_{\mathrm{d}} \right)^2 - (4AB)} \right)}{2B} \tag{2}$$

where $F_{\mathrm{n}}$ is the observed relative change in fluorescence, $F_{\mathrm{hot}}$ is the fluorescence after IR laser activation, $F_{\mathrm{cold}}$ is the fluorescence before IR laser activation, $F_{\mathrm{B}}$ is the relative change in fluorescence exhibited by the unbound labeled molecule (PHD2), $F_{\mathrm{AB}}$ is the relative change in fluorescence exhibited by the complex (PHD2/HIFα-CODD), A is the concentration of the unlabeled molecule (HIFα-CODD), B is the concentration the labeled molecule (PHD2) and $K_{\mathrm{d}}$ is the dissociation constant. In this type of experiment $F_{\mathrm{B}}$, $F_{\mathrm{AB}}$ and $K_{\mathrm{d}}$ are fitted values. Binding curves were baseline subtracted and normalized to fraction bound for Figs. 1b, c and 2b using the values for $F_{\mathrm{B}}$ and $F_{\mathrm{AB}}$, respectively. For peptides exhibiting no binding the $F_{\mathrm{AB}}$ value determined from the WT HIF2α-CODD peptide curve was used for normalization.

## Crystallographic structure determination

Purified PHD2 in Dialysis Buffer was concentrated in a 10 kDa cutoff centrifugal filter to a concentration slightly above 2 mM and supplemented with FeO$_4$ and NOG to final concentrations of 2.4 and 4 mM, respectively. The resulting solution was adjusted to a PHD2 concentration of 2 mM and incubated at room temperature for 10 min. After incubation, the 2 mM PHD2 solution was combined 1:1 with a 2.4 mM solution of WT HIF2α-CODD peptide dissolved in 50 mM Tris–HCl pH 8.0 creating a solution with 1 mM PHD2 and 1.2 mM WT HIF2α-CODD peptide an the sample was incubated at room temperature for 10 min. This solution was further diluted to a final concentration of 0.67 mM PHD2 and 0.8 mM WT HIF2α-CODD in a Dialysis Buffer prior to sparse-matrix crystallization screening using a Douglas Instruments Oryx 8 liquid handling robot. Initial crystallization hits were imaged using a Formulatrix UV Rock Imager 1000 and further optimized by varying the protein and precipitant concentrations. The best diffracting crystals were obtained via hanging drop vapor diffusion in a condition containing 32.5% (w/v) PEG 2000 MME at 20 °C. The protein crystals nucleated off a skin that formed on the drop surface and took approximately 1 month to grow to full size. Crystals were cryoprotected by 10 s sequential transfers to a solution first containing 50 mM Tris–HCl pH 7.5, 33% (w/v) PEG 2000 MME, and 15% (v/v) glycerol, then to 50 mM Tris–HCl pH 7.5, 33% (w/v) PEG 2000 MME and 30% (v/v) glycerol. Crystals were then immediately flash-cooled in liquid nitrogen.

Data were collected at the NSLS-II synchrotron beamline 17-ID-2 (FMX) at a wavelength of 0.97933 Å using a Detectris Eiger X 16M detector[33]. DIALS[34] was used to index, scale, and merge crystallographic data. The structure was determined using molecular replacement and the atomic coordinates from PDB: 5L9B[16] (PHD2/HIF1α-CODD complex) as a search model. All HIF1α-CODD and ligand atoms were removed from the search model prior to molecular replacement. The CCP4[35] software package was used to perform molecular replacement (PHASER[36]), automated model building (BUCCANEER[37]), and automated refinement (REFMAC5[38]). The complete model was built with iterative rounds of manual model building in Coot[39] followed by coordinate refinement in REFMAC. Riding hydrogens were used during refinement to improve the structure. MOLPROBITY[40] and Coot were used to perform validation of the atomic model. Protein interfaces were analyzed using PISA[41]. Structural representations and rmsd calculations were performed using PyMol. Phenix[42] was used to calculate the composite omit electron density map and to extract and compile the crystallographic statistics presented in Table 3. The B-factor modified Z-score of HIFα-CODD residues was calculated using B-factors of all C$_\alpha$ atoms in the

asymmetric unit of the respective structures[43,44]:

$$MAD = \text{median}\left( \sqrt{\left( B(i) - B_{median} \right)^2} \right) \quad (3)$$

$$B'(i) = \begin{cases} \dfrac{B(i) - B_{median}}{\frac{1.235}{N} \sum_{i=1}^{N} \left( B(i) - B_{median} \right)^2}, & MAD = 0 \\ \\ \dfrac{B(i) - B_{median}}{1.486(MAD)} & MAD \neq 0 \end{cases} \quad (4)$$

where MAD is the median absolute deviation; $B$ is the atomic $B$-factor for atom $i$; $B_{median}$ is the median $B$-factor; $B'$ is the $B$-factor modified $Z$-score of atom $i$; $N$ is the total number of $C_\alpha$ atoms.

### Dual-luciferase reporter assay

HEK293A cells were maintained in DMEM (Wisent #319-015) supplemented with 10% (v/v) Fetal Bovine Serum (FBS, Wisent #098-150) in a humidified atmosphere at 37 °C and 5% $CO_2$. Transfection complexes containing 2 µg of total DNA (0.8 ug pcDNA3-HA, 0.9 µg pGL3-VEGFa-HRE, 0.1 µg pRL-SV40, and 0.2 µg of either pcDNA3-HA-HIF2α-WT or mutant) and 8 ug Polyethyleneimine pH 7.2 (Polysciences, Cat #23966) in 400 µl OptiMEM™ Reduced Serum Medium (Gibco #31985070) were incubated for 15 min at room temperature, then were added to $5 \times 10^5$ HEK293A cells (diluted in 400 µl OptiMEM™) and incubated for an additional 5 min. Mixture was plated in six-well plates containing 2 ml complete medium and incubated at 37 °C, 5% $CO_2$ for 24 h. Firefly luciferase and Renilla luciferase activity were measured with the Dual-Luciferase Reporter Assay System (Promega #E1960) according to the manufacturer's instructions. Measurements (1 s) were recorded using a Varioskan Lux microplate reader (Thermo Fisher Scientific). For each sample, firefly RLU values were divided by corresponding Renilla RLU values.

### Reporting summary

Further information on research design is available in the Nature Portfolio Reporting Summary linked to this article.

### Data availability

The PHD2/HIF2α-CODD crystal structure has been deposited in the Protein Data Bank (PDB entry: 7UJV). The source data can be found in Supplementary Data Set 1. Numerical data used to create the plots in this work is published alongside the online version of this manuscript as a supplemental file. Uncropped blot images are provided in Supplementary Fig. 6. All other data are available from the corresponding author on reasonable request.

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

## Acknowledgements

We thank the members of the Lee and Ohh labs for their helpful comments. We thank Dr. Trevor Moraes (University of Toronto, Department of Biochemistry) for the use of the MST instrument. This work was supported by grants from the Canadian Institutes of Health Research (PJT-159773 to M.O.) and the Canada Research Chair program (CRC-2017-00140 to J.E.L.). Biophysics and structural biology infrastructure were supported by funding from the Canada Foundation for Innovation-John R. Evans Leaders Fund to J.E.L. We thank the staff on Beamline 17-ID-2 (FMX) at the National Synchrotron Light Source II (NSLS-II) for synchrotron access and support. Support for work performed at the Center for Biomolecular Structure beamline LIX (16ID)|AMX (17ID-1)|FMX (17ID-2) at NSLS-II is provided by NIGMS-1P30GM133893 and BER-BO 070. NSLS-II is supported by DOE, BES-FWP-PS001.

## Author contributions

F.G.F. designed and performed the experiments, interpreted the data, and wrote the manuscript. C.C.T. performed the in vitro hydroxylation and

binding assays. S.S. Performed the Dual-Luciferase reporter assays. M.H. assisted in the production of HIF2α expression plasmids. D.T. provided the preliminary optimization for the co-crystallization work. J.E.L. assisted in the crystallography, interpreted the data, wrote and edited the manuscript. M.O. conceptualized the project, interpreted the data, wrote and edited the manuscript. All authors reviewed and edited the manuscript.

## Competing interests

The authors declare no competing interests.
