## [Peer review file · Communications Biology]

Reviewers' comments:

Reviewer #1 (Remarks to the Author):

In this manuscript, authors used MST to determine and compare the binding affinities (K_d values) between various HIF-2 α CODD peptides (wild-type or Pacak-Zhuang-associated mutants) and the catalytic domain of PHD2. They found that PHD2 has lower affinities towards Class 1 mutants than Class 2 mutants. Consistent with the MST data, co-crystal structure of PHD2/HIF-2 α CODD peptide complex revealed that the mutated residues from Class 1 are mainly localized near the critical interface between HIF-2 α and PHD2, while those mutated residues from Class 2 are localized to the more flexible region of HIF-2 α . Finally, authors demonstrated that Class 1 mutations but not Class 2 mutations, could effectively increase HIF-2 α -mediated transcriptional activity within cells. This work reveals that the phenotype of Pacak-Zhuang syndrome is closely related to the strength of direct protein interactions between HIF-2 α CODD and PHD2, implying a potential clinical significance. However, there are still several points to be addressed by authors in their revision:

1. This study focuses on only PHD2 among three known PHD isoforms. How about the role of PHD1 or PHD3 in the regulation of HIF-2 α ? Is there a huge difference or selectivity in terms of their interactions to HIF-2 α CODD? Authors are encouraged to at least comment on these questions in the Introduction or Discussion Section.

2. In Figure 6, mutation of both prolines (P405A&P531A) exhibits a similar effect as single mutation of CODD proline (P531A). However, it would be more informative if the single mutation of NODD proline (P405A) is also presented for a direct comparison.

As in this figure, the normalized activity of HIF-2 α WT is only 2 times higher than that of Empty Vector. Since transfection of full-length HIF-2 α into cells can usually improve HIF-2 α protein level and thus greatly improve the luciferase signal, 2-fold seems to be unexpected low. Authors may try to optimize their transfection system and repeat this experiment.

3. In Abstract Section (Line 44), the cell-based experiments measuring transcriptional activities should be considered as *in vitro* rather than *in vivo* experiments.

Reviewer #2 (Remarks to the Author):

Oxygen sensing using the Hypoxia-Inducible Factor (HIF) system requires the hydroxylation of proline residues within the constitutively expressed HIF α by Prolyl Hydroxylase Domain-containing protein (PHD) during oxygenated conditions, and subsequent recognition of the hydroxylated HIF α by von Hippel-Lindau protein (VHL), leading to HIF α proteasomal degradation. In hypoxic conditions, PHD activity drops, VHL does not bind HIF α , and HIF α is stabilized promoting hypoxia response gene expression. Pathogenic variants that disrupt the oxygen sensing process and lead to disease of variable severity are known for each member of this pathway.

Here, the authors have studied the interaction between PHD2 catalytic domain and twenty pathogenic variants on the CODD region of HIF-2 α , divided into two classes based on disease severity. This work extends previous work by the group reported in reference 4. The authors first used an indirect hydroxylation assay that measures the pull-down of VHL by HIF-2 α CODD peptides after in vitro hydroxylation by PHD2. This method was found to give highly variable results that did not always correlate with the disease class of the variant in question. They then used microscale thermophoresis to determine the binding affinities of the HIF-2 α variant peptides to the PHD2 catalytic domain complexed with N-oxalylglycine and Fe²⁺. These results mostly correlated well with the disease class. The authors also report some tests comparing the binding of HIF-1 α CODD and HIF-2 α CODD to PHD2.

The authors have determined the crystal structure of the complex between PHD2 and HIF2 α -CODD peptide. Protein Data Bank contains two additional released entries of this complex, but these are currently unaccompanied by a publication. The analysis of the crystal structure focuses on the differences between this structure and the PHD2 – HIF-1 α complex structure. Authors then use the structure and attempt to understand the different binding affinities of the peptide variants and the correlated disease severity. They find that the variants linked to more severe disease occur at residues interacting directly with PHD2, while those linked with milder symptoms tend to occur at a more flexible region of the substrate disrupting its conformation. Finally, authors measured transcriptional activation from eight pathogenic HIF-2 α variants using a cellular luciferase assay.

I think the manuscript presents interesting work and would be a valuable contribution to the hypoxia field. The PHD2 – HIF-2 α crystal structure elucidates the details of this interaction that differ subtly from those of the PHD2 – HIF-1 α complex. The presented affinity assay results seem to correlate quite well with the disease severity classification. Perhaps the main shortcoming of this work, in the category of “what I would have done”, is that the authors did not use a direct method to determine the PHD2 activity towards the substrate peptides. The differences in substrate K_D towards the enzyme typically reflect differences in activity, but I would have liked to see a direct assay to determine this. Also, as the substrate peptide length is known to influence PHD activity parameters, so that longer peptides suggest tighter binding (for example PMID: 16885164), it would have been interesting to see the impact of longer peptide(s) or even full-length HIF-2 α on K_D , but perhaps that is not central for this study. I find the overall quality and presentation of the work is good. I have some specific minor points listed below that I would like the authors to correct or comment on, but other than that I think the manuscript could be published as it stands.

1. Row 151: G537W is said to be class 1 while elsewhere it is class 2.

2. Row 164-165: this and figure 2a suggest that aspartate and glutamate are “synonymous” in the context of HIF-1 α and HIF-2 α CODD sequences. I disagree. In fact, one of the pathogenic variants studied here is D539E, suggesting the two amino acids, in general, are not equal. Please, clarify in text.

3. Row 280-281: Referring to salt bridge interaction by E538, please include a mention of the other interacting residue.

4. Row 374-376: “This result suggests...”. It seems to me this is backwards. Typically, HIF α proline hydroxylation is thought to prevent transcription downstream, not to be necessary for it. Please comment or modify.

5. Discussion: The authors should comment on, if “the disruptions to the PHD2/HIF2 α -CDD interface and therefore prolyl hydroxylation via PHD2 is arguably the most influential step in the observed HIF2 α stabilization in Pacak-Zhuang syndrome.”, then why the Y532H variant with K_D essentially identical to WT is still pathogenic.
6. Row 478: Y532 should probably be Y532C.
7. Row 617: “In panels e and f,…” should probably be panels d and e.
8. Row 637: “In panels a-c,…” should probably include panel g.
9. Table 1: It is difficult for me in the printed version to differentiate between blue and purple in the HIF-2 α sequence. Some other shade of purple should be used. In addition, I think the hydroxylated proline should be highlighted in some way in the HIF-2 α sequence.
10. Table 2: Do the multiple letters signifying Pacak-Zhuang sub-classes refer to different patients? It might be useful to include in the table the number of patients that have been discovered with each mutation, if that number is available.
11. Figure 5d: Y532 is labeled as Y531.

RESPONSE TO REVIEWERS

REVIEWER No. 1 (Remarks to the Author):

In this manuscript, authors used MST to determine and compare the binding affinities (Kd values) between various HIF-2 α CODD peptides (wild-type or Pacak-Zhuang-associated mutants) and the catalytic domain of PHD2. They found that PHD2 has lower affinities towards Class 1 mutants than Class 2 mutants. Consistent with the MST data, co-crystal structure of PHD2/HIF-2 α CODD peptide complex revealed that the mutated residues from Class 1 are mainly localized near the critical interface between HIF-2 α and PHD2, while those mutated residues from Class 2 are localized to the more flexible region of HIF-2 α . Finally, authors demonstrated that Class 1 mutations but not Class 2 mutations, could effectively increase HIF-2 α -mediated transcriptional activity within cells. This work reveals that the phenotype of Pacak-Zhuang syndrome is closely related to the strength of direct protein interactions between HIF-2 α CODD and PHD2, implying a potential clinical significance. However, there are still several points to be addressed by authors in their revision:

1. This study focuses on only PHD2 among three known PHD isoforms. How about the role of PHD1 or PHD3 in the regulation of HIF-2 α ? Is there a huge difference or selectivity in terms of their interactions to HIF-2 α CODD? Authors are encouraged to at least comment on these questions in the Introduction or Discussion Section.

Response:

Done. We have expanded the discussion on page 9 (highlighted yellow, lines 340 -> 347) accordingly. Briefly, there are three PHD paralogs in humans (PHD1-3) with PHD2 being the most abundantly expressed in many cell types examined to date. Notably, many (>95) germline PHD2 mutations have been reported to be causative of erythrocytosis with or without PPGL. PHD1 and PHD3 do not appear to play an entirely redundant role with PHD2 as no known erythrocytosis- or PPGL-associated mutations have been identified in PHD3 and only 1 case has been reported with a PHD1 mutation. Thus, the deregulation of PHD2-mediated HIF2 α hydroxylation appears to be critical in the pathogenesis of Pacak-Zhuang syndrome.

2. In Figure 6, mutation of both prolines (P405A&P531A) exhibits a similar effect as single mutation of CODD proline (P531A). However, it would be more informative if the single mutation of NODD proline (P405A) is also presented for a direct comparison. As in this figure, the normalized activity of HIF-2 α WT is only 2 times higher than that of Empty Vector. Since transfection of full-length HIF-2 α into cells can usually improve HIF-2 α protein level and thus greatly improve the luciferase signal, 2-fold seems to be unexpectedly low. Authors may try to optimize their transfection system and repeat this experiment.

Response:

Done. We agree with the reviewer's suggestion and the experiment has been repeated with the addition of P405A mutant (New Fig. 6b), which did not alter the conclusion. The text in results and discussion has been updated accordingly (highlighted yellow) on pages 9 and 10 (lines 327- 328 and 380 – 386) respectively. Briefly, HIF2 α P405A/P531A double mutant displayed indistinguishable transcriptional activity from HIF2 α P531A single mutant under normoxic conditions while the P405A mutant displayed transcriptional activity comparable to wild-type HIF2 α suggesting that the elimination of P405 hydroxylation within the N-terminal ODD offers negligible activation of HIF2 α .

We also performed the same experiment with increasing amounts of plasmids encoding HIF2 α to determine whether the increased ectopic HIF2 α protein expression influenced the outcome of the experiment. While we did observe that increasing the amount of transfected HIF2 α plasmid (for example, comparing 0.2 to 1 μ g) enhanced the observed reporter activity relative to cells transfected

with empty vector (from approximately 2 to 5-fold), the corresponding fold changes in activity between wild-type HIF2 α and hydroxylation deficient mutants (P405A, P531A and P405A/P531A) remained similar to our previous measurements.

3. In Abstract Section (line 44), the cell-based experiments measuring transcriptional activities should be considered as *in vitro* rather than *in vivo* experiments.

Response:

Done. In the revised abstract, ‘*in vivo*’ has been changed to ‘*in cellulo*’ (line 44).

--

REVIEWER No. 2 (Remarks to the Author):

Oxygen sensing using the Hypoxia-Inducible Factor (HIF) system requires the hydroxylation of proline residues within the constitutively expressed HIF α by Prolyl Hydroxylase Domain-containing protein (PHD) during oxygenated conditions, and subsequent recognition of the hydroxylated HIF α by von Hippel-Lindau protein (VHL), leading to HIF α proteasomal degradation. In hypoxic conditions, PHD activity drops, VHL does not bind HIF α , and HIF α is stabilized promoting hypoxia response gene expression. Pathogenic variants that disrupt the oxygen sensing process and lead to disease of variable severity are known for each member of this pathway.

Here, the authors have studied the interaction between PHD2 catalytic domain and twenty pathogenic variants on the CODD region of HIF-2 α , divided into two classes based on disease severity. This work extends previous work by the group reported in reference 4. The authors first used an indirect hydroxylation assay that measures the pull-down of VHL by HIF-2 α CODD peptides after *in vitro*

hydroxylation by PHD2. This method was found to give highly variable results that did not always correlate with the disease class of the variant in question. They then used microscale thermophoresis to determine the binding affinities of the HIF-2 α variant peptides to the PHD2 catalytic domain complexed with N-oxalylglycine and Fe²⁺. These results mostly correlated well with the disease class. The authors also report some tests comparing the binding of HIF-1 α CODD and HIF-2 α CODD to PHD2. The authors have determined the crystal structure of the complex between PHD2 and HIF2 α -CODD peptide. Protein Data Bank contains two additional released entries of this complex, but these are currently unaccompanied by a publication. The analysis of the crystal structure focuses on the differences between this structure and the PHD2 – HIF-1 α complex structure. Authors then use the structure and attempt to understand the different binding affinities of the peptide variants and the correlated disease severity. They find that the variants linked to more severe disease occur at residues interacting directly with PHD2, while those linked with milder symptoms tend to occur at a more flexible region of the substrate disrupting its conformation. Finally, authors measured transcriptional activation from eight pathogenic HIF-2 α variants using a cellular luciferase assay.

I think the manuscript presents interesting work and would be a valuable contribution to the hypoxia field. The PHD2 – HIF-2 α crystal structure elucidates the details of this interaction that differ subtly from those of the PHD2 – HIF-1 α complex. The presented affinity assay results seem to correlate quite well with the disease severity classification. Perhaps the main shortcoming of this work, in the category of “what I would have done”, is that the authors did not use a direct method to determine the PHD2 activity towards the substrate peptides. The differences in substrate K_D towards the enzyme typically reflect differences in activity, but I would have liked to see a direct assay to determine this. Also, as the substrate peptide length is known to influence PHD activity parameters, so that longer peptides suggest tighter binding (for example PMID: 16885164), it would have been interesting to see the impact of longer peptide(s) or even full-length HIF-2 α on K_D, but perhaps that is not central for this study. I find the overall quality and presentation of the work is good. I have some specific minor points listed below that I would like the authors to correct or comment on, but other than that I think the manuscript could be published as it stands.

Response:

We thank the reviewer for their overall positive assessment. Regarding a direct method of determining PHD2 activity on substrate peptides, we are in fact in the process of developing for the first time an NMR-based approach of quantifying the rate of prolyl-hydroxylation on HIF α peptides in real-time. This is another major line of investigation that is, with all due respect, beyond the scope of this manuscript. Regarding the impact of peptide length on K_D, this is indeed not central for the present study. Focusing on the affinities of the frequently mutated region of HIF2 α CODD was sufficient for assessing the impact of each mutation relative to the wild-type counterpart sequence of equal length. As noted by the reviewer, previous data from the literature, as well as preliminary data from our own group (See Figure below), showed that the affinity of PHD2 for the entire ODD domain is higher than that of the short 20aa CODD segment examined in the present work. These findings suggest that the interaction interface between PHD2 and HIF2 α is perhaps incompletely understood, and if the number of pathogenic mutations beyond the ODD domain increases, we may need to adjust the current approach to incorporate longer peptides or regions of HIF2 α to fully understand the molecular mechanisms of disease.

During the time of our revision, the two additional PDB entries on HIF2 α CODD-PHD2 complex that were noted by the reviewer were published by the Schofield group (Figg et al., 2023 Proteins), which we have appropriately cited in our revised manuscript. Notably, their structural work does not investigate Pacak-Zhuang syndrome.

1. Row 151: G537W is said to be class 1 while elsewhere it is class 2.

Response:

Corrected. We thank the reviewer for noting this oversight. G537W is now correctly described and highlighted yellow as Class 2 on page 5 (line 151).

2. Row 164-165: this and figure 2a suggest that aspartate and glutamate are “synonymous” in the context of HIF-1 α and HIF-2 α CODD sequences. I disagree. In fact, one of the pathogenic variants studied here is D539E, suggesting the two amino acids, in general, are not equal. Please, clarify in text.

Response:

Done. We have clarified in the text that there are 4 amino acid differences between the two sequences, and that we focused on examining the two sites which we predicted would have the biggest impact on the binding interface. The change is highlighted yellow on page 5 (lines 165-168). We also adjusted Figure 2a to highlight the 4 amino acid differences in the sequence between HIF1 α and HIF2 α CODD.

3. Row 280-281: Referring to salt bridge interaction by E538, please include a mention of the other interacting residue.

Response:

Done. We have clarified in the text that HIF2 α E538 is directed towards PHD2 K297. The clarification is highlighted yellow on page 8 (line 283-284).

4. Row 374-376: “This result suggests...”. It seems to me this is backwards. Typically, HIF α proline hydroxylation is thought to prevent transcription downstream, not to be necessary for it. Please comment or modify.

Response:

Done. We have clarified in the text that P405 does not seem to be required for the suppression of HIF2 α activity since P405A transcriptional activity is comparable to wild-type HIF2 α activity and P405A/P531A activity is similar to P531A activity. The clarification is highlighted yellow on page 10 (lines 380 - 386).

5. Discussion: The authors should comment on, if “the disruptions to the PHD2/HIF2 α -CODD interface and therefore prolyl hydroxylation via PHD2 is arguably the most influential step in the observed HIF2 α

stabilization in Pacak-Zhuang syndrome.”, then why the Y532H variant with K_D essentially identical to WT is still pathogenic.

Response:

We thank the reviewer for this astute comment. We have further extended and clarified in the discussion that although PHD2-mediated hydroxylation seems to be the most influential step in the observed HIF2 α stabilization in Pacak-Zhuang syndrome, it is not the sole deregulated process as evidenced by HIF2 α Y532H mutant with near-identical affinity to PHD2 as wild-type HIF2 α , as well as the importance of subsequent pVHL recognition of HIF2 α . The change is highlighted yellow on page 10 (Lines 394-398).

6. Row 478: Y532 should probably be Y532C.

Response:

Agreed. Y532 has been corrected to Y532C (highlighted yellow on page 13) (line 489).

7. Row 617: “In panels e and f,...” should probably be panels d and e.

Response:

Done. Thank you for noting the oversight. The change is highlighted yellow on Figure 4 legend on page 16 (line 628).

8. Row 637: “In panels a-c,...” should probably include panel g.

Response:

Done. The Figure 5 legend has been corrected and highlighted yellow on page 17 (line 648).

9. Table 1: It is difficult for me in the printed version to differentiate between blue and purple in the HIF-2 α sequence. Some other shade of purple should be used. In addition, I think the hydroxylated proline should be highlighted in some way in the HIF-2 α sequence.

Response:

Done. The purple color has been adjusted to increase visibility in Table 1 and an asterisk has been added beneath P531 to indicate the hydroxylated proline residue in the sequence.

10. Table 2: Do the multiple letters signifying Pacak-Zhuang sub-classes refer to different patients? It might be useful to include in the table the number of patients that have been discovered with each mutation, if that number is available.

Response:

Done. We have modified Table 2 to indicate the number of reported patients with each mutation and specified how many individuals exhibit each sub-class phenotype. Citations for each report have also been included.

11. Figure 5d: Y532 is labeled as Y531.

Response:

Corrected. Thank you for noting the oversight.

REVIEWERS' COMMENTS:

Reviewer #1 (Remarks to the Author):

All my previous concerns have been addressed properly by the authors. I would like to congratulate them on this nice work.

Reviewer #2 (Remarks to the Author):

I find the revised manuscript has fully addressed my comments.